

# Symmetry energy in holographic QCD

Lorenzo Bartolini and Sven Bjarke Gudnason[*]

Institute of Contemporary Mathematics, School of Mathematics and Statistics,
Henan University, Kaifeng, Henan 475004, P. R. China

[*] gudnason@henu.edu.cn

## Abstract

We study the symmetry energy (SE), an important quantity in nuclear physics, in the Witten-Sakai-Sugimoto model and in a much simpler hard-wall model of holographic QCD. The SE is the energy contribution to the nucleus due to having an unequal number of neutrons and protons. Using a homogeneous Ansatz representing smeared instantons and quantizing their isospin, we extract the SE and the proton fraction assuming charge neutrality and beta-equilibrium, using quantization of the isospin zeromode. We also show the equivalence between our method adapted from solitons and the usual way of the isospin controlled by a chemical potential at the holographic boundary. We find that the SE can be well described in the WSS model if we allow for a larger 't Hooft coupling and lower Kaluza-Klein scale than is normally used in phenomenological fits.



# 1 Introduction

The equation of state (EOS) of nuclear matter is central to nuclear physics from neutron stars to heavy ion collisions, and an important feature is the symmetry energy (SE) as a function of the density. The symmetry energy is the symmetric increase in energy as one moves away from the isospin symmetric point, that is, the point where the number of protons equals that of neutrons, i.e. $E(\rho) = E_0(\rho) + S(\rho)\beta^2 + \cdots$, with $\beta = \frac{N-Z}{A}$ being the difference between the number of neutrons $N$ and the number of protons $Z$, normalized by the atomic mass number, $A = Z + N$ — for a nice review see Ref. [1]. The symmetry energy is experimentally well constrained around saturation density, $\rho_0 \sim 0.16 \text{fm}^{-3}$, to be near $S(\rho_0) \sim 30$ MeV – both from astrophysical observations as well as heavy ion collision data – but much less so at larger densities. The symmetry energy around saturation density is conventionally expanded as

$$S(\rho) = S_0 + \frac{1}{3}L\epsilon + \frac{1}{18}K_{\text{sym}}\epsilon^2 + \cdots, \tag{1}$$

with $\epsilon := (\rho - \rho_0)/\rho_0$, whereas $L$ and $K_{\text{sym}}$ are proportional to the slope and second derivative of the SE with respect to the density. Expectedly, the constraints on $L$ and $K_{\text{sym}}$ are less tight than those on $S_0$. Traditionally, the symmetry energy was defined for nuclear matter, which can be thought of as an infinitely large nucleus at density $\rho$, and so surface effects are absent. The symmetry energy can equally well be defined for a fixed, but finite, atomic number $A$.

Current experimental bounds on the first 3 observables of the symmetry energy as in the expansion of the density come from mass, radius and tidal deformation of neutron stars, excitation energies of isobaric analog states, neutron skin in Sn isotopes and $^{208}$Pb as well as heavy ion collision data [2–5, 5–9].

The equation of state in nuclear physics relates the energy density with the pressure and is the main ingredient in the understanding of neutron stars as well as heavy ion collisions. The problem with obtaining the equation of state for nuclei is that the strong nuclear force is governed by Quantum Chromodynamics (QCD), an inherently strongly coupled theory and hence cannot be tackled by perturbation theory or first-principles calculations. Nuclear physics, in particular, ab initio methods, like the no-core shell model [10], utilize pion scattering data to reconstruct the interaction potential of nuclei and this approach leads to solid predictions for the interaction potential and the chiral effective field theory can accurately determine the EOS [11], albeit only at relatively small densities. QCD at high energies is perturbative due to its asymptotically free nature, and hence can be used to make solid predictions for the EOS [12], unfortunately at pressures far larger than those of a neutron star – the most compact object known, not collapsed into a black hole (BH).

A new paradigm of studying QCD and attempting to extract observables for nuclear physics and hadronic physics, was envisioned by Maldacena at the end of the '90-ies [13] and further elaborated by Witten [14]. After a couple of decades, the mentioned framework known as holography or AdS/CFT, has been coined holographic QCD (HQCD) when applied to the strong nuclear force [15–17]. There are two main approaches to HQCD, top-down and bottom-up; the top-down approach is based on string-theory constructions and the most prominent example is the Witten-Sakai-Sugimoto model (WSS) [14, 18, 19]. For the bottom-up construction, which shares similar theoretical ingredients, two main types of models are known as soft-wall (SW) (e.g. Improved HQCD [20, 21] and V-QCD [22–24]) and hard-wall (HW) models [25–33]. Especially, the top-down type of HQCD have quite some predictive power, in the sense that the models have very few adjustable parameters [15–17]. For the WSS, there is the mass scale and the 't Hooft coupling, where the mass scale is normally fitted to the mass of the $\rho$ meson and the 't Hooft coupling is determined from the pion decay constant [18].

Attempts have already been made at extracting the SE from various HQCD models, including top-down approaches as in the D4/D6 model [34] and even the WSS model, see Ref. [35],

and NSs have been constructed using the holographically extracted EOS to solve the governing Tolman-Oppenheimer-Volkov (TOV) equations [36, 37]. The SE, however, comes out too large in the WSS [35]. In HQCD in contrast to traditional nuclear physics, the proton and the neutron are not point particles, but are described by a topological soliton, the Sakai-Sugimoto soliton [18, 38–40], which is initially isospin symmetric – that is, the proton is equal to the neutron. In order to compute the SE, we must distinguish the proton from the neutron and this can be done by the introduction of an isospin chemical potential [35].

HQCD at finite isospin chemical potential has been object of inspection in the context of many models: top-down approaches include the D3/D7 model [41–43] and the WSS model [44, 45] in the case of the DBI action and without employing the homogeneous Ansatz (see below).

In this paper, we propose using the homogeneous Ansatz in the WSS, but quantizing the isospin symmetry – a technique known from the Skyrme model [46–48], which is the leading-order low-energy effective theory of the WSS model [18]. The homogeneous Ansatz represents an approximation to describe densely packed nucleons that form nuclear matter above saturation density. It relies on the assumption that nuclear matter forms a spatially homogeneous distribution, in which nucleons lose their individual properties: despite being shown in Ref. [49] that such a configuration is not admitted in holographic models under assumptions of regularity of the gauge fields, the Ansatz can still be employed with modifications, such as formulating it at the level of the field strengths [50] or (as we will do) introducing a discontinuity that acts as a source of baryon number [51]. The quantization of the isospin symmetry introduces the isospin quantum number, which makes it possible to extract the SE as the coefficient of the square of the difference between the number of protons and neutrons. The quantization method of introducing isospin is also shown to be equivalent to using a chemical potential, see Appendix A. We find a lower SE compared to previous attempts in the WSS [35], since we include all the needed fields in our Ansatz and because we choose a different $N_c$ scaling such that the nucleons are states with the minimal isospin quantum number; however, there is no difference coming from using either the chemical potential or the quantization method – they are equivalent as shown explicitly in Appendix A. In particular, we find a phenomenologically viable value of the constant $S(\rho_0)$ at saturation density and the first two coefficients $L$ and $K_{\rm sym}$ are compatible with current experimental bounds from astrophysics and heavy ion collision data for a certain choice of the model parameters.

## 2 Model

We will treat the WSS and the HW model on equal footing in the following. The model at low energies is described by the Yang-Mills (YM) and Chern-Simons (CS) actions in 5-dimensional AdS$_5$ (or AdS$_5$-like, for the WSS model) spacetime $(M, g)$:

$$
S_{\rm YM} = -\kappa \, {\rm Tr} \int_M \mathcal{F} \wedge * \mathcal{F} \,,
$$
$$
S_{\rm CS} = \frac{N_c}{24\pi^2} \, {\rm Tr} \int_M \left[ \mathcal{A} \wedge \mathcal{F}^2 - \frac{\rm i}{2} \mathcal{A}^3 \wedge \mathcal{F} + \frac{1}{10} \mathcal{A}^5 \right] \,,
\tag{2}
$$

with $S = S_{\rm YM} + S_{\rm CS}$ being the total action, the constant $\kappa = \frac{\lambda N_c}{216\pi^3}$ for the WSS model and $\kappa = M_5$ for the HW model, $\lambda = g_{\rm YM}^2 N_c$ the 't Hooft coupling, $N_c$ the number of colors of QCD (i.e. 3 in nature), the field strength 2-form is

$$
\mathcal{F} = \tfrac{1}{2}(\partial_\alpha \mathcal{A}_\beta - \partial_\beta \mathcal{A}_\alpha + {\rm i}[\mathcal{A}_\alpha, \mathcal{A}_\beta]){\rm d}x^\alpha \wedge {\rm d}x^\beta \,,
\tag{3}
$$

$\alpha, \beta = 0, 1, 2, 3, 4$ with $x^4 = z$ the holographic coordinate, and the power of forms is understood with the wedge product. The metric is

$$g = h(z)k(z)\mathrm{d}x_\mu \mathrm{d}x^\mu + h^2(z)\mathrm{d}z^2 \,, \tag{4}$$

with $h(z) = k^{-1/3}(z) = (1 + z^2)^{-1/3}$, $z \in (-\infty, \infty)$ for WSS and $h(z) = k(z) = \mathsf{L}/z$, $z \in [0, \mathsf{L}]$ for the HW model, the index $\mu = 0, 1, 2, 3$ is summed over in the metric and $\mu$ is raised with the Minkowski metric here. It is convenient to work in dimensionless units. Both models have a free mass scale $M_{\mathrm{KK}}$ (WSS) and $\mathsf{L}^{-1}$ (HW): the WSS model is already presented with the choice $M_{\mathrm{KK}} = 1$, to which we add the choice $\mathsf{L} = 1$ for the HW model. The correct powers of the energy scales $M_{\mathrm{KK}}$, $\mathsf{L}^{-1}$ can then easily be restored via dimensional analysis.

The gauge field can be decomposed for later convenience in the Abelian and non-Abelian parts as

$$\mathcal{A}_\alpha = A_\alpha^a T^a + \widehat{A}_\alpha \frac{\mathbb{1}}{2} \,, \tag{5}$$

where the generators of SU(2) $T^a$ are chosen as $T^a = \frac{1}{2}\tau^a$ so that $\mathrm{Tr}\, T^a T^b = \frac{1}{2}\delta^{ab}$, and the spacetime indices follow the convention:

$$\alpha, \beta, \ldots = \{0, M\} \,, \qquad M, N, \ldots = \{i, z\} \,, \qquad \mu, \nu, \ldots = \{0, i\} \,. \tag{6}$$

In writing Eq. (2), we performed dimensional reduction in the WSS, integrating out $S^4$ from the original nine-dimensional action for the stack of $D8-$Branes, while in the HW we do not explicitly include an action for the scalar field encoding chiral symmetry breaking, since we set the scalar field to zero, which is appropriate in the homogeneous baryonic phase, following Ref. [37]. Despite not appearing explicitly in our computation, the scalar field plays an important role: its vacuum energy, determines the density of nuclear matter at the baryonic onset, hence defining saturation density within this model. For details on how the scalar field defines the saturation density, but otherwise vanishes in the baryonic phase, see Appendix D. Here we will utilize the fit found in Ref. [37] and only adjust the overall energy scale.

Two further steps were employed in order to write the action and equations of motion for the two models in a compact way. For the HW model we assumed the symmetry properties for the fundamental fields $\mathcal{L}_M, \mathcal{R}_M$ as follows:

$$\mathcal{L}_i = -\mathcal{R}_i \,, \qquad \mathcal{L}_0 = \mathcal{R}_0 \,. \tag{7}$$

For the WSS model, we similarly assumed parity properties of the fields with respect to $z$:

$$\mathcal{A}_i(z) = -\mathcal{A}_i(-z) \,, \qquad \mathcal{A}_0(z) = +\mathcal{A}_0(-z) \,. \tag{8}$$

With these procedures, we halve the number of fields in the HW model (from $\mathcal{L}, \mathcal{R}$ to $\mathcal{A}$) and the integration interval in the WSS model (from $(-\infty, +\infty)$ to $[0, +\infty)$) generating an overall factor of 2 in the action in both cases.

As a last step, we introduce generic symbols $z_{\mathrm{IR}}, z_{\mathrm{UV}}$ to indicate the infrared and ultraviolet boundary values of the holographic coordinate,[1] which in the two models assume the values

$$z_{\mathrm{IR}} = \begin{cases} 0 \,, & \mathrm{WSS} \,, \\ 1 \,, & \mathrm{HW} \,, \end{cases} \qquad z_{\mathrm{UV}} = \begin{cases} +\infty \,, & \mathrm{WSS} \,, \\ 0 \,, & \mathrm{HW} \,. \end{cases} \tag{9}$$

---

[1]In the WSS model, the spatial manifold only has a UV boundary at $z = \pm\infty$. However, when we introduced the "folding" of the coordinate $z$ exploiting the assumptions (8), we effectively introduced an IR boundary at the folding point $z_{\mathrm{IR}} = 0$. Moreover, the homogeneous Ansatz will introduce a discontinuity in the field $A_i$ at that point.

The classical homogeneous Ansatz for isospin-symmetric matter, reasonable for large-density computations, is defined as

$$\mathcal{A}_0^{\text{cl}} = \tfrac{1}{2}\widehat{a}_0\,, \qquad \mathcal{A}_i^{\text{cl}} = -\tfrac{1}{2}H\tau^i\,, \qquad \mathcal{A}_z^{\text{cl}} = 0\,, \tag{10}$$

where $\{\widehat{a}_0, H\} = \{\widehat{a}_0, H\}(z)$ are functions of the holographic coordinate $z$. We have suppressed the unit 2-by-2 matrices in the terms without a Pauli matrix $\tau$.

The function $H(z)$ encodes the baryonic density through its value at $z = z_{\text{IR}}$: if either $H(z_{\text{IR}})$ or $H'(z_{\text{IR}})$ vanish, then the baryon number would also vanish as noted in Ref. [49], so we will assume that $H(z)$ obeys a Dirichlet boundary condition $H(z_{\text{IR}}) = H_0$, with the value of $H_0$ to be determined by minimization of the action. This defines the baryon density $\rho$ (assuming $H(z) \to 0$ for $z \to z_{\text{UV}}$) as follows:

$$
\begin{aligned}
\rho &= \frac{1}{16\pi^2}\int \mathrm{d}z\, \epsilon^{MNPQ}\,\text{Tr}\,F_{MN}F_{PQ} \\
&= -\frac{3}{4\pi^2}\int \mathrm{d}z\, H'H^2 \\
&= -\epsilon\frac{1}{4\pi^2}\big[H^3\big]_{z_{\text{IR}}}^{z_{\text{UV}}}\,,
\end{aligned}
\tag{11}
$$

so that the infrared boundary condition for the numerical integration of the function $H(z)$ is directly related to the baryon number density as:

$$H(z_{IR}) = \epsilon\left(4\pi^2\rho\right)^{\frac{1}{3}}\,, \tag{12}$$

where for convenience of putting the two models on same footing, we have defined the integral in the holographic direction as

$$\int \mathrm{d}z\, f(z) := \epsilon \int_{z_{\text{IR}}}^{z_{\text{UV}}} \mathrm{d}z\, f(z)\,, \tag{13}$$

which we will use throughout the paper and $\epsilon$ assumes a different sign depending on the model:

$$\epsilon = \begin{cases} +1\,, & \text{WSS}\,, \\ -1\,, & \text{HW}\,. \end{cases} \tag{14}$$

Thus the integral is defined in such a way to take into account the different orientation in the integration along $z$, dictated by the choice of coordinates for the two models. Note that this choice of boundary condition for $H(z_{\text{IR}})$ means that in the WSS, once we restore the full domain of integration $z \in (-\infty, \infty)$, the function $H(z)$ will be discontinuous. This still leads to a continuous field strength, since both $H'$ and $H^2$ are continuous functions. For the HW model instead, this choice just means that we cannot enforce the standard boundary condition $L_\mu(z_{\text{IR}}) - R_\mu(z_{\text{IR}}) = 0$, which has to be replaced with the one above, implying $L_\mu(z_{\text{IR}}) = -R_\mu(z_{\text{IR}})$.

## 3 Time-dependent configurations

We wish to include the effects of isospin asymmetry in the system. To do so, we follow a method inspired by the single-soliton analysis: we know that for the single baryon, the proton and the neutron are described as degenerate (in absence of quark mass terms[2]) quantum

---

[2]See Ref. [52] for the effect of breaking the degeneracy for the WSS model, when including the quark mass terms. For some recent results regarding nuclear matter that include quark masses in the WSS model, see Refs. [53–55]. For results on the phase diagram of a HW model including the effects of quark mass, see Ref. [56].

states of the effective Hamiltonian obtained by considering a slow rotation in SU(2). The homogeneous Ansatz shares a similar structure with the single-soliton configuration, made easier by the absence of translational moduli[3] $X_M$ and $\rho$ (but with the minor complication of not having an analytical configuration to approximate our static Ansatz (10)), so we can attempt to follow steps similar to the ones in Refs. [38] and [39], in order to obtain a time-dependent configuration – yet to be quantized.

We start by assuming a configuration of the form

$$A_0 = 0 \,, \tag{15}$$

$$A_i = V A_i^{\mathrm{cl}} V^{-1} - i V \partial_i V^{-1} \,, \tag{16}$$

$$A_z = -i V \partial_z V^{-1} \,, \tag{17}$$

which implies the following transformations in the field strengths:

$$F_{MN} = V F_{MN}^{\mathrm{cl}} V^{-1} \,, \tag{18}$$

$$F_{0z} = -V D_z^{\mathrm{cl}} \Phi V^{-1} \,, \tag{19}$$

$$F_{0i} = 0 \,, \tag{20}$$

where $V(z, t)$ encodes the time-dependent rotation in SU(2), and $\Phi$ is defined as

$$\Phi \equiv -i V^{-1} \dot{V} \,. \tag{21}$$

Notice, this is not a gauge transformation since the field $A_0$ is not transformed along with the rest. The function $V(z, t)$ needs to depend on $z$ in order to allow us to satisfy the equation of motion

$$-\kappa \left( h(z) D_j F^{0j} + D_z \left( k(z) F^{0z} \right) \right) + \frac{N_c}{64\pi^2} \epsilon^{0\alpha_1\alpha_2\alpha_3\alpha_4} F_{\alpha_1\alpha_2} \widehat{F}_{\alpha_3\alpha_4} = 0 \,. \tag{22}$$

The function $V(z, t)$ is holographically dual to the SU(2)-valued collective coordinate $a(t)$, as we choose it such that

$$V(z \to z_{\mathrm{UV}}, t) = a(t) \,, \tag{23}$$

which in turn implies

$$\Phi(z \to z_{\mathrm{UV}}, t) = -i a^{-1} \dot{a} \equiv \frac{1}{2} \boldsymbol{\chi} \cdot \boldsymbol{\tau} \,, \tag{24}$$

where $\boldsymbol{\chi}$ is the boundary angular velocity. The presence of a nonvanishing $F_{0z}$ will also enable a source term for the fields $\widehat{A}_i$ via the Chern-Simons action, so we will have to complete the field content by turning on $\widehat{A}_i = -\frac{1}{2} L \chi^i$: here we already guessed that the vector field will be proportional to the angular velocity $\chi^i$, and we can do so without loss of generality, since in the homogeneous case this is the only three-vector available to the Abelian field.

At this stage the problem is well posed and the function $\Phi(z, t)$ can be found by solving Eq. (22), but it is more convenient to perform a gauge transformation to make the system easier to treat.

We perform the gauge SU(2) transformation

$$A_\alpha \to A_\alpha^S = G A_\alpha G^{-1} - i G \partial_\alpha G^{-1} \,, \qquad G \equiv a V^{-1} \,, \quad \alpha = 0, 1, 2, 3, 4 \,, \tag{25}$$

where the superscript "$S$" stands for "singular", because this is reminiscent of the transformation changing from the regular gauge to the singular gauge in the single-soliton case. With this

---

[3]The translational moduli $X_i$ are absent because of the assumption of homogeneity, while the pseudo-modulus size $\rho$ is fixed by the numerical solution so as to minimize energy. The pseudo-modulus $Z$ describing the center of the soliton in $z$ is fixed by our Ansatz to be at the position of the discontinuity. This in principle can also be determined by choosing $Z = z_0$ that minimizes the free energy as opposed to our simpler choice $z_0 = 0$ for all densities. See Ref. [57] for the inclusion of this effect in the static approximation.

choice (dropping the superscript "$S$" for convenience, since we will use this gauge henceforth) the field content becomes

$$A_0 = a\left(\Phi - \frac{1}{2}\tau \cdot \chi\right)a^{-1}, \tag{26}$$

$$A_i = aA_i^{\text{cl}}a^{-1}, \tag{27}$$

$$A_z = 0. \tag{28}$$

Now we can factorize the function $\Phi(z,t)$ as

$$\Phi = \Phi^a \chi^a \equiv \widetilde{G}\chi \cdot \tau, \tag{29}$$

and since we imposed Eq. (24), we see that

$$\widetilde{G}(z \to z_{\text{UV}}) = \frac{1}{2}. \tag{30}$$

We then conclude that the field $A_0$ in this gauge vanishes at the UV boundary, and can be expressed as

$$A_0 = G(z)a\chi \cdot \tau a^{-1}, \qquad G(z \to z_{\text{UV}}) = 0. \tag{31}$$

We notice that this result is exactly what one would expect by allowing for the most general field configuration respecting spherical symmetry, homogeneity in three-dimensional flat space, and the gauge choice $A_z = 0$. Taking the functions $H, \widehat{a}_0, G, L$ to be independent of $\chi$ amounts to considering a slow rotation, thus including only linear terms in $\chi$ in the Ansatz.

Whereas $\frac{1}{2}\chi \cdot \tau$ is the matrix form of the boundary angular velocity, $\frac{1}{2}a\chi \cdot \tau a^{-1} = -i\dot{a}a^{-1}$ is the matrix form of the boundary angular isospin velocity (i.e. describing rotations in SU(2) instead of in space). Thus, although one may think we are spinning the fields in space, this is really an isospin action on the homogeneous fields.

The final form of our time-dependent homogeneous Ansatz is then summarized in compact notation as:

$$\mathcal{A}_0 = Ga\chi \cdot \tau a^{-1} + \frac{1}{2}\widehat{a}_0, \qquad \mathcal{A}_i = -\frac{1}{2}\left(H\tau^i a^{-1} + L\chi^i\right), \qquad \mathcal{A}_z = 0, \tag{32}$$

with the mandatory boundary condition $G(z \to z_{\text{UV}}) = 0$.

This Ansatz leads to the action

$$S_{\text{YM}} = -\kappa \int \mathrm{d}^4 x \int \mathrm{d}z \Bigg[ -8hH^2\left(G + \frac{1}{2}\right)^2 \chi \cdot \chi + 3hH^4$$
$$+ k\left[(L')^2 - 4(G')^2 + 8(KH)^2\right]\chi \cdot \chi + 3k(H')^2 - k(\widehat{a}_0')^2 \Bigg], \tag{33}$$

$$S_{\text{CS}} = -\frac{N_c}{8\pi^2}\int \mathrm{d}^4 x \int \mathrm{d}z\, \widehat{a}_0 H'H^2 + \frac{N_c}{4\pi^2}\int \mathrm{d}^4 x \int \mathrm{d}z\left(LH' - L'GH\right)H\chi \cdot \chi, \tag{34}$$

which gives rise to the equations of motion

$$hH^3 - \frac{1}{2}\partial_z(kH') - \frac{N_c}{32\pi^2\kappa}H^2\widehat{a}_0' = 0, \tag{35}$$

$$\partial_z(k\widehat{a}_0') + \frac{3N_c}{16\pi^2\kappa}H^2H' = 0, \tag{36}$$

$$\partial_z(kG') - hH^2(1 + 2G) + \frac{N_c}{32\pi^2\kappa}H^2L' = 0, \tag{37}$$

$$\partial_z(kL') + \frac{N_c}{8\pi^2\kappa}H\left[HG' + (1 + 2G)H'\right] = 0, \tag{38}$$

where the first two equations of motion are truncated to order $|\chi|^0$, whereas the latter two only appear at quadratic order in $\chi$. Including the subleading $|\chi|^2$ corrections to the solutions of $H$ and $\widehat{a}_0$ has a negligible impact, which we checked explicitly. Moreover, in the limit of small $\chi$, quadratic corrections in $\chi$ to the functions $H(z), \widehat{a}_0(z)$ would generate terms in the on-shell action at order $|\chi|^4$, not contributing to the symmetry energy, and also being subleading in the small $\chi$ expansion.

This set of equations is composed by ODEs in the holographic coordinate $z$ and can be solved with standard off-the-peg solvers in packages like MATHEMATICA or MATLAB, once we specify all the boundary conditions:

$$G'(z_{\text{IR}}) = -\frac{N_c}{32\pi^2\kappa}H^2(z_{\text{IR}})L(z_{\text{IR}}), \qquad \widehat{a}_0'(z_{\text{IR}}) = L'(z_{\text{IR}}) = 0, \qquad H(z_{\text{IR}}) = \epsilon(4\pi^2\rho)^{\frac{1}{3}}, \quad (39)$$

and all fields are vanishing at $z = z_{\text{UV}}$. These boundary conditions are obtained by imposing the vanishing of also the total derivative that comes about when deriving the Euler-Lagrange field equations; for more details, see Ref. [58].[4] We recall that that in the chosen coordinates $z_{\text{IR}} = 0$ ($z_{\text{IR}} = 1$), $z_{\text{UV}} = \infty$ ($z_{\text{UV}} = 0$) and $\epsilon = +1$ ($\epsilon = -1$) for the WSS (HW) model.

Another possible approach would be to keep the fields in a static configuration, hence keeping the freedom to set the standard orientation of Eq. (10), and introduce an external isospin chemical potential, which holographically amounts to introducing a finite UV boundary value for the field $A_0$: in Appendix A we show that this approach is related to ours by a gauge transformation, hence leading to the same physics. This formalism is the one employed in [35, 55]: the two calculations, however, differ in that in the present work we have turned on the Abelian field $\widehat{A}_i$, which turns out to be linear in $\chi$, and we are effectively truncating the $\chi$ dependence of the gauge fields at linear order. The inclusion of the Abelian component is necessary to have a self-consistent Ansatz, as the equations of motion cannot be solved by setting $L(z) = 0$ (the Chern-Simons term provides a source for $L(z)$). Moreover, it turns out that the field $L(z)$ dominates the small-$\lambda$ behavior of the symmetry energy: despite the holographic model being developed with the large-$\lambda$ limit in mind, for the practical application of extracting a value for the symmetry energy, we need to extrapolate to a finite-$\lambda$, and the most popular fit of the model employs the value of $\lambda = 16.63$, which does not realize the large-$\lambda$ nor the small-$\lambda$ regimes (see Appendix E). On top of the difference at the level of the Ansatz, another difference with respect to Refs. [35, 55] lies in the implicit definition of the isospin number of nucleon states in the large-$N_c$ limit. We choose as proton (neutron) state the lowest-lying isospin state, which can be thought of as being composed of $\frac{1}{2}(N_c + 1)$ up (down) and $\frac{1}{2}(N_c - 1)$ down (up) quarks. With this definition, the angular velocity $\chi$ of a nucleon state is of order $N_c^{-1}$, and so are the isospin chemical potential and the symmetry energy.[5] A different choice that still reduces to the familiar $N_c = 3$ case is that in which the proton (neutron) is composed of $N_c - 1$ up (down) and one down (up) quarks: in this scenario the isospin number is of order $N_c$, and so is the symmetry energy. We find appropriate the former definition for nucleon states, in that it keeps the nucleons as the ground state baryons, and preserves the familiar electric charge following the Gell-Mann-Nishijima formula

$$Q = I_3 + \frac{N_B}{2}, \quad (40)$$

with $Q, I_3, N_B$ being the electric charge, the third component of the isospin, and the baryon number, respectively.

---

[4]Imposing the coupled Robin-type boundary condition for $G'$ as opposed to a Neumann boundary condition (a naive but consistent choice based on the field's parity if $L(z_{\text{IR}}) = 0$) only leads to a decrease in the symmetry energy of about 10-20%.

[5]Note that the symmetry energy is a $1/N_c^2$ correction to the leading $\mathcal{O}(N_c)$ baryon energy, while corrections from the axial anomaly would be further suppressed as $1/N_c$ and can provide corrections to the symmetry energy only at order $\mathcal{O}(N_c^{-2})$, see Appendix C.

The truncation of the $\boldsymbol{\chi}$ dependence (and so of the dependence on $\mu_I$) is an approximation that does not affect the computation, since the symmetry energy is by definition obtained by evaluating the first nonvanishing term in the expansion of the energy per nucleon around an isospin symmetric configuration.

Despite this technique being equivalent to the usual introduction of a boundary chemical potential $\mu_I$, it has a series of advantages, particularly manifest in the small $\mu_I$ limit. In this limit, the complicated picture of the isospin asymmetric homogeneous matter becomes similar to the well understood one of a slowly rotating bulk instanton, and the problem of finding the symmetry energy becomes the computation of a moment of inertia.

It is also built-in in this formalism what the smallest value of the isospin is. Identifying this smallest unit of isospin with that of a single quark being flipped from down to up fixes the isospin quantum number without $N_c$-scaling ambiguities.

On top of this simplification, our alternative technique also helps in identifying the solution to the problem of the ambiguity of the Chern-Simons term when dealing with homogeneous nuclear matter. As pointed out in Ref. [58], boundary terms arising from the Chern-Simons term can contribute with an IR effective action because of the discontinuity of $H(z)$. In this case, different choices for the Chern-Simons action that differ by a boundary term are not physically equivalent (as opposed to the case of a smooth instanton), as they enforce different IR boundary conditions on the flavor fields, with the consistent case giving rise to the boundary conditions (39).

A way of solving the ambiguity is to require that the holographic currents on the boundary match with their sources in the bulk: in Ref. [58] this was done for the baryonic density $\rho$ (requiring that the topological charge matches with the baryonic current in the tail of $\hat{a}_0$) and for the isospin density $\rho_I$ (requiring that the angular velocity description matches with the isospin chemical potential one).

This kind of problem arises in every holographic model containing a Chern-Simons term in the action, hence this result obtained with the aid of the new quantization technique is very generalizable and will prove useful in identifying the correct Chern-Simons action for future works exploring isospin asymmetry in holographic models. The correct choice of the action is crucial to obtain the correct thermodynamic quantities, so an improvement in this regard directly translates to a more precise equation of state, and more reliable predictions for properties of neutron stars.

## 4 Symmetry energy

The terms quadratic in $\boldsymbol{\chi}$ exactly produce the SE upon Hamiltonian quantization:

$$
\begin{aligned}
H &= \frac{1}{2} V \Lambda \boldsymbol{\chi} \cdot \boldsymbol{\chi} + V U \\
&= 2 V \Lambda \dot{a}_m^2 + V U \\
&= \frac{\pi_m^2}{8 V \Lambda} + V U \\
&= \frac{I(I+1)}{2 V \Lambda} + V U \,,
\end{aligned}
\tag{41}
$$

where canonical quantization of $a_m$, $m = 0, 1, 2, 3$, a coordinate on the 3-sphere ($a_m^2 = 1$), leads to the momentum conjugate

$$
\pi_m = \frac{\partial H}{\partial \dot{a}_m} = 4 V \Lambda \dot{a}_m \,,
\tag{42}
$$

and hence to $\pi_m^2 = \ell(\ell + 2)$ being the spherical harmonics and $\ell = 2I$, with $I$ the isospin quantum number [46].[6] The identification of $V\chi^2$ and $I(I+1)/V$ coming from Hamiltonian quantization is also justified by the holographic dictionary, since it can be obtained by computing the third component of the isovector charge density, see Appendix B for a detailed computation. The functionals $\Lambda$ and $U$ are defined as

$$
\Lambda = 2\kappa \int dz \left[ 2hH^2(2G+1)^2 + k((L')^2 + 4(G')^2) \right],
$$
$$
U = \kappa \int dz \left[ 3hH^4 + 3k(H')^2 + k(\widehat{a}_0')^2 \right],
$$
(43)

where $V$ denotes the spatial 3-volume. Using now the relation between isospin and the number of protons and neutrons:

$$
2I = Z - N = -\beta A,
$$
(44)

with $Z$ the proton number and $N$ the neutron number, as well as the atomic number

$$
A = Z + N = V\rho,
$$
(45)

being the product of the 3-volume and the baryonic density. $\beta$ is defined as the normalized difference between the number of neutrons and protons, $\beta = (N-Z)/A$, hence we have

$$
\frac{H}{A} = \frac{U}{\rho} + S(\rho)\beta^2 + \mathcal{O}(V^{-1}),
$$
(46)

$$
S(\rho) = \frac{\rho}{8\Lambda},
$$
(47)

where $S(\rho)$ is the symmetry energy as a function of the density.

Using the standard phenomenological fit for the WSS model of Ref. [19], we set $\lambda = 16.63$ and find the first SE expansion parameters as

$$
S_0 = 74.9 \left( \frac{M_{\text{KK}}}{949\,\text{MeV}} \right) \text{MeV},
$$
$$
L = 113.3 \left( \frac{M_{\text{KK}}}{949\,\text{MeV}} \right) \text{MeV},
$$
(48)
$$
K_{\text{sym}} = -35.9 \left( \frac{M_{\text{KK}}}{949\,\text{MeV}} \right) \text{MeV},
$$

which are somewhat larger than values typically obtained from phenomenological models [1], but much smaller than obtained in the WSS previously [35]. For the HW model, we fix $M_5 = \frac{N_c}{12\pi^2}$ using the leading OPE coefficient of the vector current correlator [30], for which the first few SE expansion parameters are

$$
S_0 = 70.4 \left( \frac{L^{-1}}{150\,\text{MeV}} \right) \text{MeV},
$$
$$
L = 132.5 \left( \frac{L^{-1}}{150\,\text{MeV}} \right) \text{MeV},
$$
(49)
$$
K_{\text{sym}} = -218.8 \left( \frac{L^{-1}}{150\,\text{MeV}} \right) \text{MeV},
$$

---

[6]Due to the simplicity of the homogeneous Ansatz, the isospin quantum number is identical to the spin quantum number in magnitude; this is an artifact of the Ansatz, but it does not increase the kinetic energy. In particular, for reading off the coefficient of the symmetry energy at $\beta = 0$, this artifact of the approximation of using the homogeneous Ansatz is irrelevant.

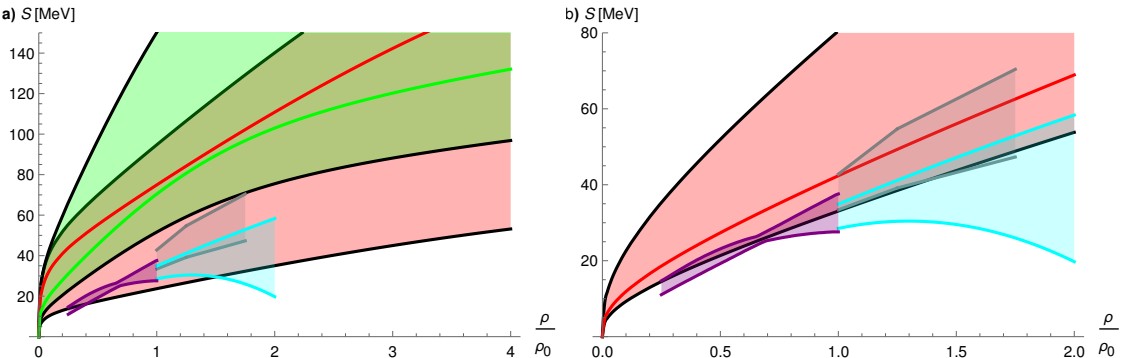

Figure 1: The symmetry energy (SE) calculated in the WSS model with the phenomenological value of the 't Hooft coupling and in the HW model, both using quantization of isospin as functions of the density. a) The red area corresponds to the WSS model with $M_{KK}$ ranging from 300 MeV to 1200 MeV and the red line in the middle is at 949 MeV. The green area corresponds to the HW model with $L^{-1}$ ranging from 110 MeV to 320 MeV and the green line in the middle is at 150 MeV. The constraints from the PREX-II experiment using the neutron skin thickness of $^{208}$Pb [6] are shown with a gray shaded area, while the extensive 2021 survey of constraints on the symmetry energy of Li et.al. [4] using neutron stars, are shown with a cyan shaded area. Constraints from isobaric analog states below saturation density are shown with a purple shaded area [2]. b) The SE calculated in the WSS model with the 't Hooft coupling $\lambda = 60$. The red shaded area corresponds to $M_{KK} \in [390, 949]$ MeV and the red solid curve is the rescaled phenomenological mass scale, that keeps the pion decay constant at 93 MeV, corresponding to $M_{KK} = 500$ MeV.

where $L^{-1}$ is the mass scale of the HW model, which we set as $L^{-1} = 150$ MeV following Ref. [37], which provides phenomenologically good results for neutron stars in terms of mass-radius data. The saturation density, $\rho_0$, in HQCD is defined to be at the onset of baryonic matter, obtained by minimization of the free energy with the baryonic chemical potential as the boundary condition $\hat{a}_0(z_{UV}) = \mu_B$. The value of $\rho_0$ obtained this way in the WSS model with $\lambda = 16.63$ is

$$\rho_0 = 0.436 \left( \frac{M_{KK}}{949 \, \text{MeV}} \right)^3 \text{fm}^{-3} \,, \tag{50}$$

which is about 2.9 times too large with respect to the phenomenological value – an overestimate by the same order of magnitude as the other baryonic quantities. The HW model, however, yields a more realistic saturation density

$$\rho_0 = 0.183 \left( \frac{L^{-1}}{150 \, \text{MeV}} \right)^3 \text{fm}^{-3} \,, \tag{51}$$

which is only about 22% too large.

We explore a larger range of densities for both the WSS and the HW model in Fig. 1a). For the WSS model, we have used the phenomenological value of the 't Hooft coupling ($\lambda = 16.63$) and shown the range of $M_{KK} \in [300, 1200]$ MeV with a red shaded area, which includes $M_{KK} = 949$ MeV [19] (red solid line), whereas for the HW model the range of $L^{-1} \in [110, 320]$ MeV is shown with a green shaded area, which includes $L^{-1} = 150$ MeV (green solid line) that is chosen from neutron star phenomenology [37] and the highest mass scale is from meson physics [31, 59]. Up-to-date constraints from astrophysics and heavy-ion collision data are shown with gray and cyan shaded areas near and above saturation density and constraints using nuclear excitation energies from isobaric analog states (IAS) are shown

with a purple shaded area below saturation density. As can be seen from the figure, the phenomenologically fitted value of the mass scale $M_{\text{KK}}$ at 949 MeV [38] overestimates the SE with about a factor of 2.4; however, the fit is made using mesonic observables and is known to overestimate baryonic observables; for instance, the baryon mass is typically overestimated by a factor of 1.7-1.8 [38,60,61] using the mesonic fit.

Although both models come in the ball park of the experimental constraints above saturation density and nuclear physics predictions below saturation density if we allow ourselves to adjust the energy scale ($M_{\text{KK}}$ or $L^{-1}$), the shape of the SE does not quite satisfy all the constraints. In the WSS model, however, we can dial the 't Hooft coupling to see whether we can fit in the allowed regions and indeed it is possible by raising both the 't Hooft coupling from the phenomenological value to $\lambda = 60$, as well as lowering the KK scale from 949 MeV to 390 MeV, see Fig. 1b); this corresponds to the lower black curve of the red shaded area. With the larger 't Hooft coupling, the SE of the WSS has a compatible shape to pass all the constraints, but the $\rho$ meson is too light and the baryon mass is too heavy – often a problem in HQCD; the baryon mass is reduced from about 1600 MeV to 1191 MeV. If we keep the pion decay constant at its phenomenological value, the KK scale, however, is lowered too and is shown in Fig. 1b) with a solid red curve – not too far from a viable solution. Recomputing the symmetry energy expansion parameters, we obtain

$$
\begin{aligned}
S_0 &= 33.1 \left( \frac{M_{\text{KK}}}{390\,\text{MeV}} \right) \text{MeV}, \\
L &= 66.4 \left( \frac{M_{\text{KK}}}{390\,\text{MeV}} \right) \text{MeV}, \\
K_{\text{sym}} &= -34.3 \left( \frac{M_{\text{KK}}}{390\,\text{MeV}} \right) \text{MeV},
\end{aligned}
\tag{52}
$$

which are compatible with phenomenological constraints. The saturation density for this fit now also has improved as

$$
\rho_0 = 0.166 \left( \frac{M_{\text{KK}}}{390\,\text{MeV}} \right)^3 \text{fm}^{-3},
\tag{53}
$$

which is only about 10% from the phenomenological value. Lowering the KK scale from 390 to 380 MeV will improve both $S_0$ and $\rho_0$ ($S_0 = 32.2$ MeV and $\rho_0 = 0.153$ fm$^{-3}$), but will create a bit more tension with the constraint coming from the neutron skin thickness of $^{208}$Pb, shown with a gray-shaded area in Fig. 1b). A similar choice of fit is employed in Ref. [62], where we choose $M_{\text{KK}}$ and $\lambda$ in order to reproduce the correct saturation density and SE. The resulting equation of state (hybridized with phenomenological low-density EOSs) then is found to reproduce viable properties of neutron stars.

It is not surprising that a somewhat larger value of $\lambda$ is needed in order to more reliably reproduce the physics of baryonic matter. We can understand it by considering the single baryon in the WSS model: with the BPST instanton approximation it is possible to compute both the classical mass and its quantum corrections [38]. In particular, since we are interested in the symmetry energy, we want to consider the (iso)spin quantum correction, given by integer values of $l$ in the formula [38]

$$
M = 8\pi^2 \kappa + \sqrt{\frac{(l+1)^2}{6} + \frac{2}{15} N_c^2} + \frac{2(n_\rho + n_Z + 1)}{\sqrt{6}},
\tag{54}
$$

where $n_\rho, n_Z$ are quantum numbers for the size and bulk position excitations.

It is well known that the usual mesonic fit with $M_{\text{KK}} = 949$ MeV, $\lambda = 16.63$ largely overestimates the nucleon masses, but a major contribution in this result comes from the fact that the quantum corrections are close in magnitude to the classical mass. We can see that the classical

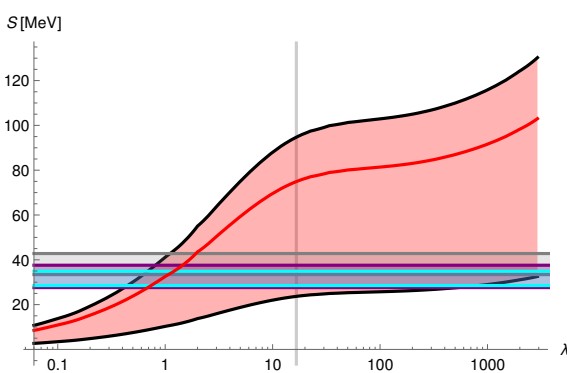

Figure 2: The symmetry energy (SE) calculated in the WSS model using quantization of isospin as a function of the 't Hooft coupling $\lambda$ at saturation density. The red area spans the mass scale $M_{KK}$ from 300 MeV to 1200 MeV and the red line is at 949 MeV. The gray, cyan and purple shaded areas are the same as in Fig. 1. The vertical gray line marks the phenomenological 't Hooft coupling $\lambda = 16.63$ [19].

mass is of order $\mathcal{O}(\lambda)$, while the quantum corrections are of order $\mathcal{O}(1)$. By increasing $\lambda$ it is then possible to reduce the relative magnitude of the quantum corrections as compared to the classical result, and by reducing $M_{KK}$ it is possible to obtain an overall more realistic nucleon mass.

The same kind of mechanism is inherited by the homogeneous system, where increasing $\lambda$, the symmetry energy contribution becomes smaller, moving towards the real world value, provided that we also adjust $M_{KK}$ accordingly. While it is fairly easy to expect that some values of $\lambda$, $M_{KK}$ exist that both fit the phenomenology of saturation density and symmetry energy, it is less trivial that once these values are employed, other baryonic observables are then improved, as it happens in our case with the BPST instanton mass and with the expansion parameters $L$, $K_{sym}$.

What this analysis suggests is that when describing baryons within the WSS model, the errors introduced by the many approximations employed to make the system approachable seems to be partially mitigated by an alternative choice of the values of the parameters $\lambda$, $M_{KK}$.

The dependence on the 't Hooft coupling for the WSS model is shown in Fig. 2 for the KK scale in the interval 300-1200MeV at saturation density.

## 5 Proton fraction

We will now consider the proton fraction at $\beta$-equilibrium with charged leptons, imposing charge neutrality. Using the Gell-Mann-Nishijima formula, we can relate the baryon density, $\rho$, and isospin density, $\rho_I$, with the proton/neutron densities:

$$\rho_{P,N} = \tfrac{1}{2}\rho \pm \rho_I \,, \tag{55}$$

where the upper sign is for protons and the lower for neutrons. Charge neutrality is imposed by

$$\tfrac{1}{2}\rho + \rho_I = \sum_\ell \rho_\ell \,, \tag{56}$$

with $\ell = e, \mu$ being a sum over the charged leptons and the $\beta$-equilibrium (from the decay $N \to P + \ell + \bar{\nu}_\ell$) amounts to

$$\mu_\ell = \mu_N - \mu_P = -\mu_I \,, \quad \ell = e, \mu \,, \tag{57}$$

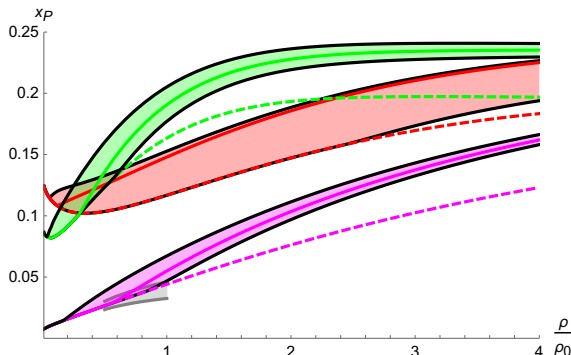

Figure 3: The proton fraction calculated in the WSS and HW model as functions of the density. The red shaded area corresponds to the WSS model with the phenomenological 't Hooft coupling $\lambda = 16.63$ and $M_{KK} \in [300, 1200]\,\text{MeV}$, the red line at $M_{KK} = 949\,\text{MeV}$ and the red dashed line at $M_{KK} \to 0$. The green shaded area corresponds to the HW model with $\mathsf{L}^{-1} \in [110, 320]\,\text{MeV}$, the green line at $\mathsf{L}^{-1} = 150\,\text{MeV}$ and the green dashed line is the limit $\mathsf{L}^{-1} \to 0$. The magenta shaded area corresponds to the WSS model with the calibration of Fig. 1b), i.e. $\lambda = 60$ and $M_{KK} \in [390, 949]\,\text{MeV}$, the solid magenta curve corresponds to the phenomenological pion decay constant, and the dashed magenta curve corresponds to $M_{KK} \to 0$, thus eliminating the muons. The gray shaded area is the result from chiral EFT [63].

where $\mu_X$ is the chemical potential of the particle species $X$. The lepton density is calculated assuming it to be a (massive) Fermi gas as [35]

$$\rho_\ell = \Theta_H(\mu_\ell - m_\ell) \frac{(\mu_\ell^2 - m_\ell^2)^{\frac{3}{2}}}{3\pi^2}, \tag{58}$$

with $\Theta_H$ being the Heaviside step function, $\mu_\ell$ the chemical potential and $m_\ell$ being the mass of the lepton $\ell$. Using the definition of the isospin chemical potential as being the conjugate variable of the isospin density, we get

$$\mu_I = \frac{1}{V}\frac{\partial H}{\partial \rho_I} = \frac{\rho_I}{\Lambda(\rho)}. \tag{59}$$

Inserting Eq. (58) into the charge neutrality condition (56) and using the $\beta$-equilibrium condition (57), we obtain an implicit solution for the isospin density, $\rho_I$, as a function of the density $\rho$:

$$\frac{\rho_I^3}{3\pi^2\Lambda^3}\left[\Theta_H(-\rho_I) + (1 - R^{-2}m_\mu^2)^{\frac{3}{2}}\Theta_H(-R - m_\mu)\right] + \rho_I + \frac{1}{2}\rho = 0, \quad R = \frac{\rho_I}{\Lambda}, \tag{60}$$

where we have set the electron mass to zero and the dimensionless muon mass parameter is the ratio of the physical mass (105.7 MeV) to the mass scale $M_{KK}$ and $\mathsf{L}^{-1}$, for the WSS and the HW models, respectively.

In Fig. 3 we show the numerical results for both the WSS and the HW model for the proton fraction at various densities around saturation density. We find that the WSS model phenomenologically fitted to mesons gives more realistic proton fractions (red shaded area) than the HW model (green shaded area) and yields even better proton fractions below saturation density if we use the calibration from Fig. 1b), i.e. $\lambda = 60$ and $M_{KK} = 390\,\text{MeV}$. Since we take the electrons to be massless, the mass scale of the model only enters in the muon mass parameter. The dashed curves correspond to the muon being infinitely heavy (or the mass scale of the model being sent to zero).

## 6 Discussion and outlook

In this paper, we have computed the symmetry energy in two holographic QCD models using the method of quantizing the isospin symmetry, namely in the top-down WSS model and the bottom-up HW model. We find fairly good agreement between our model results for the SE and proton fraction in both the HW model, using the fit from neutron stars and in the WSS model with a new calibration (i.e. $\lambda \sim 60$ and $M_{\text{KK}} \sim 390\,\text{MeV}$).

We have also shown that the method known from Skyrmions of quantizing the isospin zeromode is equivalent to introducing a chemical potential on the holographic boundary for the gauge fields, see Appendix A. There is mathematically no difference between the two methods.

It would be interesting in future work to take into account the strange quark (3 instead of 2 flavors) or alternatively the kaons, to see at what densities it might have an impact on the SE. Furthermore, there are certain transitions that happen at larger densities, for example the Skyrmion-half-Skyrmion transition [64], which has an analog in holographic instantons [65]. Although it is not directly observable in our homogeneous Ansatz, it may have some effect on the SE and proton fractions at large densities. Another approximation we employed in our calculations is that of keeping fixed the position of the discontinuity that sources the baryonic charge. It is expected that the location of the baryon is dynamically determined, moving towards the boundary as the density is increased: this behavior is the (homogeneous version of) the so-called "popcorn transition" that is known to occur in HQCD [66]. However, as shown in Ref. [37], the homogeneous Ansatz already reproduces features of the popcorn transition even when keeping the position of the discontinuity fixed in the bulk: it does so by having the baryonic holographic density form a peak at a location in the bulk that depends on the boundary density $\rho$. Because of this, we expect corrections coming from the dynamical determination of the discontinuity location to be particularly small.

## Acknowledgments

We thank Nicolas Kovensky, Anton Rebhan and Andreas Schmitt for discussions and comments on the draft.

**Funding information** The work of L. B. is supported by the National Natural Science Foundation of China (Grant No. 12150410316). S. B. G. thanks the Outstanding Talent Program of Henan University and the Ministry of Education of Henan Province for partial support. The work of S. B. G. is supported by the National Natural Science Foundation of China (Grants No. 11675223 and No. 12071111) and by the Ministry of Science and Technology of China (Grant No. G2022026021L).

## A Equivalence between rotation in SU(2) and external isospin chemical potential

We start with our field configuration given by Eq. (32), of which we rewrite the non-Abelian components:

$$A_0 = Ga\boldsymbol{\chi} \cdot \boldsymbol{\tau} a^{-1}, \tag{A.1}$$

$$A_i = -\frac{H}{2} a \tau^i a^{-1}, \tag{A.2}$$

$$A_z = 0. \tag{A.3}$$

We now perform a gauge transformation, with the aim of obtaining a static configuration in the limit of constant $\boldsymbol{\chi}$: we choose a gauge function $b(t)$ that only depends on time, so that the fields transform as

$$A_0 \to \widetilde{A}_0 = Gba\boldsymbol{\chi} \cdot \boldsymbol{\tau} a^{-1} b^{-1} - ib\partial_0 b^{-1}, \tag{A.4}$$

$$A_i \to \widetilde{A}_i = -\frac{H}{2}ba\tau^i a^{-1} b^{-1}, \tag{A.5}$$

$$A_z \to \widetilde{A}_z = 0. \tag{A.6}$$

Now we choose $b = a^{-1}$, hence rotating the fields $A_i$ back to the standard orientation, while modifying the field $A_0$ with an additional term:

$$\widetilde{A}_0 = G\boldsymbol{\chi} \cdot \boldsymbol{\tau} - ia^{-1}\dot{a}, \tag{A.7}$$

$$\widetilde{A}_i = -\frac{H}{2}\tau^i, \tag{A.8}$$

$$\widetilde{A}_z = 0. \tag{A.9}$$

We recognize the quantity of Eq. (24) in the last term of Eq. (A.7):

$$-ia^{-1}\dot{a} = \frac{1}{2}\boldsymbol{\chi} \cdot \boldsymbol{\tau}, \tag{A.10}$$

so that we are left with

$$\widetilde{A}_0 = \left(G + \frac{1}{2}\right)\boldsymbol{\chi} \cdot \boldsymbol{\tau}. \tag{A.11}$$

We know that by construction the function $G(z)$ vanishes at the boundary at $z_{\mathrm{UV}}$, so we conclude that this configuration behaves as:

$$\widetilde{A}_0(z \to z_{\mathrm{UV}}) = \frac{1}{2}\boldsymbol{\chi} \cdot \boldsymbol{\tau}. \tag{A.12}$$

The boundary value of the field $A_0$ is dual to an isospin chemical potential in the holographic dictionary. Since the orientation of the soliton is a zeromode, we can set $\boldsymbol{\chi}$ to point in a chosen direction for simplicity without loss of generality: we choose it to have only a nonvanishing third component as $\boldsymbol{\chi} = (0, 0, \mu_I)$, following the same convention of choosing the third component of isospin as the operator to diagonalize simultaneously with the isospin squared (and the isospin chemical potential to appear holographically as the boundary value of $A_0^{a=3}$). Shifting $G(z)$ as

$$\widetilde{G}(z) = \left(G(z) + \frac{1}{2}\right), \tag{A.13}$$

we obtain the familiar expressions for the gauge field and its boundary condition

$$\widetilde{A}_0 = \widetilde{G}\tau^3\mu_I, \qquad \widetilde{A}_0(z \to z_{\mathrm{UV}}) = \frac{1}{2}\mu_I\tau^3. \tag{A.14}$$

We then conclude that a static system in the presence of an external isospin chemical potential $\mu_I$ is equivalently described as it rotating in isospin space with angular velocity $\chi^i = \mu_I\delta^{i3}$, as observed in Ref. [67] in the non-holographic context of the Skyrme model.

We want to emphasize that, despite our solution to the system of coupled equations of motion is performed in the limit of small angular velocity (small $\mu_I$), the equivalence between the two methods just shown holds true in general, since we made no assumptions on the $\boldsymbol{\chi}$ dependence of the functions $H, \widehat{a}_0, G, L$. The assumption of small $\boldsymbol{\chi}$ will not affect in any way the calculation of the symmetry energy, as it is the coefficient of a term of an expansion around symmetric matter, hence all the functions would have to be evaluated at vanishing isospin

density (and $\mu_I$) anyway. For calculations at higher isospin chemical potential, the equivalence still holds, but since the now large angular velocity $\chi$ backreacts every field, this framework loses its main advantage of factorizing away the dependence on $\chi$. Moreover, the isospin symmetric Ansatz is not consistent anymore, and the function $H(z)$ has to be substituted with a set of functions $H_i(z)$ [55]:

$$A_i = -\frac{H}{2}\tau^i \qquad \Rightarrow \qquad A_i = -\frac{H_i}{2}\tau^i, \tag{A.15}$$

where $i$ is not summed over. Then the problem to be solved is that of five ($H_1(z) = H_2(z)$ because of residual symmetry) coupled ODEs, with $\chi$ dependence in each of them, effectively the same as we would have if we worked with a boundary chemical potential $\mu_I$.

Despite this not being a necessity for the calculation of the symmetry energy (since by definition it is calculated at vanishing $\mu_I$), when moving to finite isospin density (for example when computing the full phase diagram or properties of neutron stars) the more rigorous approach would be to include the effects above (and possibly non-diagonal terms for $A_i^a$ when including also quark masses, see Ref. [55] for the validity limits of the diagonal approximation), which is equally difficult with both gauge choices.

## B  Isospin density from holographic current

In this section we want to prove that the isospin number density that we defined from the quantized angular momentum coincides with the canonical one obtained through the holographic dictionary via the computation of the isovectorial current. For simplicity, we will show the proof in the WSS model, so that $z_{\mathrm{IR}} = 0$, $z_{\mathrm{UV}} = +\infty$, but it holds true in the HW model too, after substitution of the appropriate quantities. As shown in Ref. [39], the vectorial current is obtained as

$$\mathcal{J}_{V\mu} = -\kappa\left[k(z)\mathcal{F}_{\mu z}\right]_{-\infty}^{+\infty} = -2\kappa\left[k(z)\mathcal{F}_{\mu z}\right]_0^{+\infty}. \tag{B.1}$$

With this quantity we can build the isovectorial charge $Q_V$ of which we take the third component to coincide with the isospin operator

$$Q_V^{a=3} = I_3 = \int d^3x\,\mathrm{Tr}\left(J_V^0\tau^3\right) = V\,\mathrm{Tr}\left(J_V^0\tau^3\right). \tag{B.2}$$

We plug in this formula the homogeneous Ansatz (32):

$$I_3 = -2\kappa V\left[G'k(z)\right]_0^{+\infty}\chi^i\,\mathrm{Tr}\left(a\tau^i a^{-1}\tau^3\right) \tag{B.3}$$

$$= -2\kappa V\left[G'k(z)\right]_{z=+\infty}\chi^i\,\mathrm{Tr}\left(a\tau^i a^{-1}\tau^3\right), \tag{B.4}$$

where we used the fact that $G'(0) = 0$.

The angular velocity $\chi^i$ is related to the angular momentum operator $J^i$ by the familiar relation involving the moment of inertia $\Lambda$:

$$\chi^i = \frac{1}{V\Lambda}J^i, \tag{B.5}$$

and we can exploit the useful relationship between angular momentum and isospin operators that holds due to the spherical symmetry of the system:

$$J^i\,\mathrm{Tr}\left(a\tau^i a^{-1}\tau^a\right) = -2I_a, \tag{B.6}$$

so that we are left with:

$$I_3 = \frac{4\kappa}{\Lambda}\left[G'k(z)\right]_{z=+\infty}I_3. \tag{B.7}$$

We see that the validity of this relationship depends on whether the following identification holds true:

$$4\kappa \left[ G'k(z) \right]_{z=+\infty} = \Lambda \,. \tag{B.8}$$

To prove this relationship, we first notice that the formula for the current is obtained by differentiating the action with respect to the UV boundary value of the $A_0$ field, following

$$\delta_{A_0} S = (\text{e.o.m. terms}) + 4\kappa V \operatorname{Tr}\left[ k(z)A_0' \delta A_0 \right]_{z=+\infty} \,, \tag{B.9}$$

where the first term means that we neglect contributions that vanish by the equations of motion, we evaluate only boundary terms, and we made use of the boundary condition $A_0'(z = 0) = 0$. We decide to employ the gauge in which the field $A_0$ has a finite boundary value, dual to the isospin chemical potential, so we take the field to be as in Eq. (A.14). With this configuration, the variation of the action assumes the shape

$$\delta_{A_0} S = (\text{e.o.m. terms}) + 2\kappa V \operatorname{Tr}\left[ k(z)\widetilde{G}' \tau^3 \tau^3 \right]_{z=+\infty} \mu_I \delta\mu_I \,, \tag{B.10}$$

and finally we can compute the derivative

$$\frac{\partial S}{\partial \mu_I} = 4\kappa V \left[ k(z)\widetilde{G}' \right]_{z=+\infty} \mu_I \,. \tag{B.11}$$

We can change this result to our usual "rotating" gauge by noting that we have to rename $\mu_I \to \chi_3$, and that $\widetilde{G}' = G'$, so that on-shell we obtain

$$\frac{\partial S}{\partial \chi_3} = 4\kappa V \left[ k(z)G' \right]_{z=+\infty} \chi_3 \,. \tag{B.12}$$

We now look at the definition of $\Lambda$: it is nothing but the part of the energy density that is quadratic in the angular velocity, and there is no linear term. In this picture, the system is rotating and there is no chemical potential, so the on-shell action gives the energy of the system, so that we can write

$$\frac{\partial S}{\partial \chi_3} = V\Lambda\chi_3 \,. \tag{B.13}$$

Comparing Eqs. (B.11) and (B.13), we finally prove Eq. (B.8).

## C  The subleading order in $N_c$ of the chiral anomaly

Throughout the main body of this work, we have ignored the presence of the chiral anomaly of QCD: we expect on general grounds that the currents dual to the holographic fields $\mathcal{A}_\alpha$ are conserved, with the exception of the axial U(1) current, since the corresponding symmetry is broken by the chiral anomaly. Analogously, we expect the Goldstone boson associated to the axial symmetry to acquire a finite mass as a consequence of the anomaly.

This holds true in the WSS model, where the mechanism is incorporated nontrivially from the top-down construction in string theory. The model includes Ramond-Ramond forms $C_n$ of odd rank $n$: Among these is $C_7$, whose action, inclusive of a coupling with the flavor branes, reads

$$S_{C_7} = -\frac{1}{4\pi}(2\pi\ell_s)^6 \int dC_7 \wedge \star dC_7 + \frac{1}{2\pi} \int C_7 \wedge \operatorname{Tr} \mathcal{F} \wedge \omega_y \,, \tag{C.1}$$

where a one-form $\omega_y = \delta(y)dy$ has been introduced to model the distribution of the stack of branes in the $y$-direction (by definition transverse to $z$), extending the otherwise 9-dimensional integral to the whole 10-dimensional spacetime.

We can write the equation of motion as

$$\mathrm{d} \star \mathrm{d}C_7 = \mathrm{d} \star F_8 = \frac{1}{(2\pi\ell_s)^6} \operatorname{Tr}\mathcal{F} \wedge \delta(y)\mathrm{d}y\,, \tag{C.2}$$

and then use Hodge duality $\star F_8 = (2\pi\ell_s)^{-6}\widetilde{F}_2$ to turn Eq. (C.2) into an anomalous Bianchi identity:

$$\mathrm{d}\widetilde{F}_2 = \operatorname{Tr}\mathcal{F} \wedge \delta(y)\mathrm{d}y\,. \tag{C.3}$$

This form is gauge invariant if we allow $C_1$ to transform with a U(1) transformation of the flavor group:

$$\delta_\Lambda \mathrm{d}C_1 = \sqrt{\frac{N_f}{2}}\mathrm{d}\Lambda \wedge \delta(y)\mathrm{d}y\,, \qquad \delta_\Lambda \widehat{A} = -\mathrm{d}\Lambda\,. \tag{C.4}$$

The implication of this fact is that $\mathrm{d}C_1$ is not a gauge invariant form, only $\widetilde{F}_2$ is the correct gauge invariant combination.

This is welcome, since in the model $C_1$ is dual to the $\theta$ angle of QCD as

$$\theta + 2\pi k = \int_{S^4_{\mathrm{UV}}} C_1\,. \tag{C.5}$$

Let us now consider a zeromode for the field $\widehat{A}_z$, dual to the $\eta'$ meson, such that

$$\int \mathrm{d}z \widehat{A}_z = \frac{2\eta'(x)}{f_\pi}\,, \tag{C.6}$$

and plug it into the action

$$S_{\widetilde{F}_2} = -\frac{1}{4\pi(2\pi\ell_s)^6} \int \mathrm{d}^{10}x |\widetilde{F}_2|^2\,. \tag{C.7}$$

The result is an action that displays a mass term for the $\eta'$:

$$S_{\widetilde{F}_2} = -\frac{\chi_g}{2} \int \mathrm{d}^4x \left(\theta + \frac{\sqrt{2N_f}}{f_\pi}\eta'\right)^2\,, \tag{C.8}$$

with the $\eta'$ mass agreeing with the Witten-Veneziano formula

$$m^2_{\eta'} = \frac{2N_f}{f^2_\pi}\chi_g\,. \tag{C.9}$$

The topological susceptibility $\chi_g$ and the pion decay constant are computed in the model (see Ref. [18]):

$$\chi_g = \frac{\lambda^3 M^4_{\mathrm{KK}}}{4(3\pi)^6}\,, \qquad f_\pi = 2\sqrt{\frac{\kappa}{\pi}}\,. \tag{C.10}$$

We now recall that the parameter $\kappa$ was defined to be $\kappa \equiv \frac{\lambda N_c}{216\pi^3}$: this means that the mass term for the $\eta'$ meson is of order $N_c^{-1}$ (since $f^2_\pi \propto \mathcal{O}(N_c)$), while the action for the flavor fields that we employed in the main body of this work is of order $N_c$. While true that the angular velocity $\chi_i$ itself is of order $N_c^{-1}$, hence pushing the isospin asymmetric action (proportional to $\chi^2$) to be of order $N_c^{-1}$, we have to recall that eventual isospin asymmetric terms will carry factors of $N_c^{-1}$ or higher also in the $\eta'$ mass term, pushing it to even higher order in the $N_c^{-1}$ expansion. Hence it is formally safe to neglect the contribution from the axial anomaly in the large-$N_c$ scheme of approximation.

# D   The vanishing of the hard-wall tachyon in the baryonic phase

Let us quickly review the setup of the hard-wall model of Ref. [37] that utilizes a scalar field with an IR potential to dynamically stabilize it. The metric is given by

$$ds^2 = \frac{L^2}{z^2}\left(dx^\mu dx_\mu - dz^2\right), \tag{D.1}$$

where $L := 1$ is the curvature scale of $AdS_5$ and set equal to one. Upon restoring units, energies are multiplied by a physical scale.

For two flavors, we have left and right U(2) gauge fields, $\mathcal{L}_M$, $\mathcal{R}_M$ and the minimal action [37]:

$$S = S_g + S_{CS} + S_\Phi + S_{IR}, \tag{D.2}$$

$$S_g = -\frac{M_5}{2}\int d^4x dz\, a(z) \text{Tr}\left(\mathcal{L}_{MN}\mathcal{L}^{MN} + \mathcal{R}_{MN}\mathcal{R}^{MN}\right), \tag{D.3}$$

$$S_{CS} = \frac{N_c}{16\pi^2}\int d^4x dz\, \frac{1}{4}\epsilon^{MNOPQ}\left[\widehat{L}_M\left(\text{Tr}(L_{NO}L_{PQ}) + \frac{1}{6}\widehat{L}_{NO}\widehat{L}_{PQ}\right)\right.$$
$$\left. -\widehat{R}_M\left(\text{Tr}(R_{NO}R_{PQ}) + \frac{1}{6}\widehat{R}_{NO}\widehat{R}_{PQ}\right)\right], \tag{D.4}$$

$$S_\Phi = M_5\int d^4x dz\, a^3(z)\left[\text{Tr}(D_M\Phi)^\dagger(D^M\Phi) - a^2(z)M_\Phi^2\, \text{Tr}\,\Phi^\dagger\Phi\right], \tag{D.5}$$

$$S_{IR} = \frac{1}{2}m_b^2\xi^2 - \lambda_b\xi^4, \tag{D.6}$$

where $a(z) = L/z$, the U(2) gauge field $\mathcal{L}_M$ is split into SU(2) and U(1) parts as

$$\mathcal{L}_M = L_M^a\frac{\tau^a}{2} + \widehat{L}_M\frac{\mathbf{1}_2}{2}, \tag{D.7}$$

and similarly for $\mathcal{R}_M$, the field strength for $\mathcal{L}_M$ is

$$\mathcal{L}_{MN} = \partial_M\mathcal{L}_N - \partial_N\mathcal{L}_M + i[\mathcal{L}_M, \mathcal{L}_N], \tag{D.8}$$

and similarly for $\mathcal{R}_{MN}$, the covariant derivative for the scalar field is defined as

$$D_M\Phi = \partial_M\Phi + i\mathcal{L}_M\Phi - i\Phi\mathcal{R}_M, \tag{D.9}$$

the boundary condition for the scalar field is

$$\Phi(z_{IR}) = \xi\mathbf{1}_2, \tag{D.10}$$

which is stabilized by the boundary potential as the minimization of the vacuum solution and is given by

$$\xi^2 = \xi_0^2 = \frac{m_b^2 - 12M_5/L}{4\lambda_b}, \tag{D.11}$$

the mass of the scalar is $M_\Phi^2 L^2 = -3$, the would-be quark mass in the model is switched off, the indices $M, N = 0, 1, 2, 3, z$ run over all $AdS_5$, and finally $N_c$ is the number of colors and $M_5$ is a coupling of the theory (playing the role of $\kappa$ in the WSS model), which we have set as $M_5 L = N_c/(12\pi^2)$ [37].

Chiral symmetry breaking is done in Ref. [37] following [68, 69] as $(L_{z\mu} + R_{z\mu})_{z=z_{IR}} = 0$, and hence we choose $L_z = R_z = 0$ (gauge choice), $L_i = -R_i$, $\widehat{L}_0 = \widehat{R}_0$ and $\Phi$ diagonal, which means that the scalar field only couples to $L_i = -R_i$ via the covariant derivative

$$D_0\Phi = 0, \qquad D_i\Phi = \partial_i\Phi + 2iL_i\Phi, \qquad D_z\Phi = \partial_z\Phi. \tag{D.12}$$

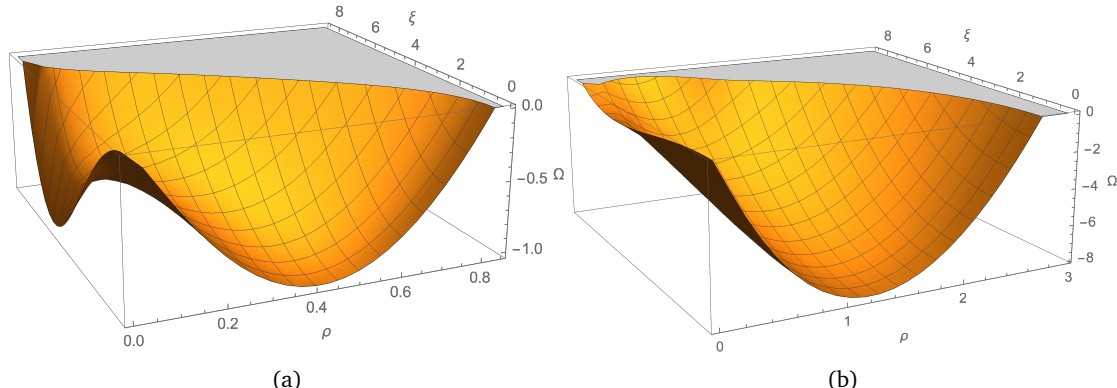

Figure 4: The action evaluated as a surface for densities $\rho$ and scalar field with coefficient $\xi$ at (a) (for (b) twice) the chemical potential that corresponds to saturation density. In this figure, the IR potential is chosen as $m_b = 0.657$ and $\lambda_b = 0.001$, giving $\lambda_b \xi_0^4 = 1.024$.

Employing the homogeneous Ansatz

$$L_i = -R_i = -H(z)\frac{\tau^i}{2}, \qquad \widehat{L}_0 = \widehat{R}_0 = \widehat{a}_0(z), \qquad \Phi = \omega_0(z)\frac{\mathbf{1}_2}{2}, \tag{D.13}$$

and the coordinates $z_{\text{UV}} = 0$, $z_{\text{IR}} = L = 1$, we can write down the vacuum solution

$$\Phi = \xi z^3 \mathbf{1}_2, \tag{D.14}$$

which holds when $H = 0$ that corresponds to $\rho = 0$ – the vanishing baryonic density and $\xi = \xi_0$ of Eq. (D.11). Once the IR potential has been fixed by choosing the two parameters $m_b$ and $\lambda_b$ that correspond to a certain $\xi_0$, the impact of the scalar is just to define the vacuum value of the action in the mesonic phase of the theory. It can readily by computed to be

$$S = -\lambda_b \xi_0^4. \tag{D.15}$$

The boundary conditions for the fields $H(z)$ and $\widehat{a}_0(z)$ are

$$H(0) = 0, \qquad H(1) = -(4\pi^2\rho)^{\frac{1}{3}}, \tag{D.16}$$

$$\widehat{a}_0(0) = \mu, \qquad \widehat{a}_0'(1) = 0, \tag{D.17}$$

with $\mu$ being the (baryonic) chemical potential.

In the phase $\rho > 0$, the vacuum of the theory is still given by $H = \widehat{a}_0 = 0$ and $\Phi$ given by the vacuum solution (D.14) until the baryonic onset, which corresponds to the nuclear saturation density. At the onset, there are two vacua: a mesonic and baryonic one each with the same value of the action (by definition), see Fig. 4(a). Once $\rho > \rho_{\text{crit}}$ one may ask at what configuration the scalar field stabilizes at. It turns out by numerical computations that the scalar turns off, which corresponds to $\xi = 0$ in the baryonic phase, see Fig. 4(b). This corresponds to $\Phi = 0$ and when studying only the baryonic phase, the impact of the scalar is to set the saturation density of this simplistic hard-wall model.

# E  Comparison with large-$\lambda$ approximation

In the large-$\lambda$ limit, one may consider ignoring the inclusion of the Abelian field $L$ (in Eq. (32)), which however is not a consistent choice for finite values of $\lambda$, as the equation of motion for $L$ is not satisfied by $L = 0$ (it is sourced by $H$ and $G$, see Eq. (38)).

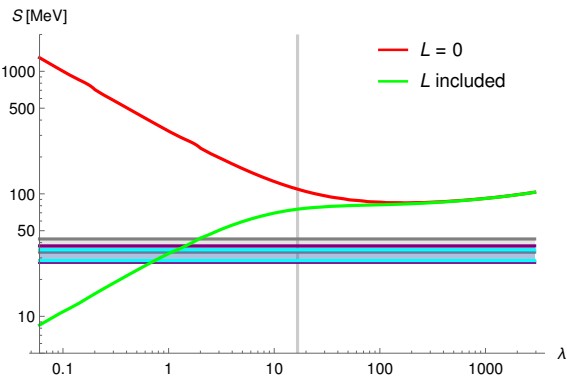

Figure 5: The symmetry energy calculated as function of $\lambda$ at saturation density in the WSS model with $L$ taken into account (green solid line) and without $L$ (i.e. $L = 0$ in the Ansatz) (red solid line). In this figure, we have used $M_{KK} = 949$ MeV.

This approximation can be seen as consistent in the large-$\lambda$ limit by considering that the field $L$ is only sourced by the Chern-Simons, which is indeed subleading in $\lambda$ with respect to the Yang-Mills terms. One may be led to believe that the same is true for the field $G$, and that they have to appear at the same order in $\lambda$, but a closer look at its equation of motion (37) shows that a source is present already in the Yang-Mills terms, coming from the time derivative of the field $A_i$ (in our time-dependent gauge, while the same contribution arises from the UV boundary condition of $\widetilde{G}$ in the static gauge).

Neglecting the Abelian field $L$ was one of the approximations made in Ref. [35] in addition to choosing a different $N_c$-scaling of the isospin chemical potential (essentially defining the large-$N_c$ baryon's proton and neutron by maximally flipping the down and up quarks, as opposed to our definition where only one quark is flipped from down to up). In order to see quantitatively how good the approximation of using $L = 0$ in the homogeneous Ansatz is, we perform the numerical calculation corresponding to Fig. 2 with and without $L$ taken into account, see Fig. 5. From the figure, we can see that the large-$\lambda$ approximation works in the sense that the two results asymptote to the same curve for $\lambda \gtrsim 200$. At $\lambda = 16.63$ and for $M_{KK} = 949$ MeV the correct computation including $L$ yields a symmetry energy of 74.9 MeV compared to 109.1 MeV when neglecting $L$ in the Ansatz, which at the meson-fitted value of $\lambda$ gives an excess in the symmetry energy of 46%. The smaller $\lambda$ is the worse it gets, as expected from a large-$\lambda$ approximation.

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
