# Peer review of "Symmetry energy in holographic QCD"

_SciPost Physics, doi:SciPost Phys. 16, 156 (2024)_

## Round 1 · Referee Report · Anonymous (Referee 1) · 2024-4-10

Strengths

The paper has a clear goal, the procedure is carefully explained, and the results obtained and comparisons with previous publications are clearly discussed.

Weaknesses

Given the equivalence with previous works discussed briefly in the main text and in detail in appendix A (up to some details) the paper would benefit from a more explicit discussion of the motivation for carrying out the present computation, not only from the technical point of view but also in view of possible future directions of investigation.

Report

In holographic QCD models baryons are usually understood as gauge instantons in the bulk. However, the difficulty in handling many instanton systems requires the use of some approximations when using this formalism to describe dense QCD-like matter. The present manuscript considers the homogeneous Ansatz routinely used in this context for the WSS and HW models, and successfully carries out an analog of the isospin quantization procedure perfomed originally for the single-instanton in the WSS model (and also in the Skyrme model). The analysis is valid at small isospin chemical potential, which is enough to compute the symmetry energy, roughly speaking around saturation density, that is, in the regime relevant for Neutron Star matter.

Appart from a few minor points, detailed below, in my opinion the paper is technically interesting, and timely, as the application of holographic methods to Neutron Star physics has led to important results in recent years. Hence, I believe it should eventually be published in SciPost Physics.

My only important suggestions for improving the final version are the following:

  • The authors mention in the introduction, main text, and show in appendix A, that the method used here is basically equivalent to that of Ref [35]. The only two differences are (1) the choice of the Nc scaling in the definition of the isospin chemical potential and the symmetry energy, and (2) the correct inclusion of the abelian spatial components, which were originally ignored in [35] (although the authors might comment on the discussion given in Ref [54] regarding this point, namely the impact of this approximation on the phase diagram of the model). However, when discussing the results for the symmetry energy and proton fraction in Secs 4 and 5, the authors say that they are much closer to the phenomenological values. I believe it should be clarified how much of this improvement actually relies on the conventional Nc scaling, and how much is due to the improvement of the Ansatz for the Abelian gauge fields.

  • Besides the computation of a more realistic symmetry energy and proton fraction (and especially if the improvement is mostly due to the Nc scaling), I believe the authors should clarify what is the main advantage of carrying out the computation in this way. For instance, what is the impact for future applications to Neutron Stars? How would those hypothetical results compare with those from other holographic NS models, such as those based on VQCD? What other applications would benefit from the quantization analysis?

  • In the discussion section, it is said that the holographic popcorn transition is "already taken into account", but this is the first mention of it. I believe this comment is too vague. If it is part of the computation, it should be explained in the main text when it enters, and it's impact should be discussed, as well as it's comparison with non-holographic models, quarkyonic matter, etc.

Requested changes

Besides addressing the questions raised in the above report, here is a list of minor points the authors should consider:

  • At the beginning of section 2, it is said that at low energies both models are described by gauge theories on AdS5, but then the metric for the WSS model is different. I understand what the authors mean, but the presentation seems a bit confusing.

  • The use of the coordinate "z" in both cases has its advantages, but it is a bit confusing that in the HW case there is a scale "L" for the AdS radius, but in the WSS model "z" is dimensionless. Perhaps this could be clarified as well.

  • In footnote 3 one could also consider citing recent papers where the effect of the quark mass on the phase diagram of the system was considered.

  • Below Eqs (35)-(38) it would be useful to explain more carefully (at least in words) why some terms quadratic in \chi are important while others are ignored.

  • It would be interesting to discuss in more detail what happens with the equivalence discussed in App. A for larger values of the isospin chemical potential, and how it could affect studies of the phase diagram and of Neutron Stars within these models.

Recommendation

Ask for minor revision

---

## Round 1 · Referee Report · Anonymous (Referee 2) · 2024-5-9

Report

The paper applies two holographic models to compute the symmetry energy of nuclear matter. In both models a simple approximation of dense, homogeneous nuclear matter is used. The questions addressed in the paper are timely and the results are new (improving on existing results in the literature). The paper is well written, all calculations are explained in a clear and transparent way. I do not have any major concerns and thus recommend publication in principle. I have some comments that should be address before publication:

(1) In the abstract, the authors claim that after adjusting the parameters the results "pass all experimental constraints". Isn't this a bit too strong? For instance, the authors say themselves in the paper that the saturation density is not reproduced correctly (and they never mention additional properties of nuclear matter such as the binding energy or the compressibility). Additionally, they also do discuss in the paper that mesonic properties and the nucleon mass are not correctly reproduced (with the new parameters used here). All this makes the formulation in the abstract very misleading and I would strongly suggest to find a more honest formulation.

(2) Somewhat related to the previous point: The authors say on p10 that the saturation density rho_0 is a factor 3 too large. But this can only be true for a fixed parameter set. It would be good to know what that value is in the regime where the symmetry energy assumes realistic values. Also, rho_0 probably changes from curve to curve in fig 1 for example. Some warning or a comment on the range of values rho_0 can assume might be helpful.

(3) A more general comment about the interpretation of the results: All numerical results in Sec 4 and 5 are presented more or less as observations without trying to interpret or understand them. For instance, do the results suggest to keep using the models because the symmetry energy comes out right, despite shortcomings for rho_0 (and possibly the never mentioned binding energy)? Or should the models or the approximations for nuclear matter be improved? Are the deviations and (partial) agreement with real-world physics expected and can they be explained for instance with large-N_c artifacts? Is there a physical explanation why a larger 't Hooft coupling leads to a more realistic symmetry energy? Given that the paper also contains interesting theoretical novelties it might be acceptable to keep these interpretations short, but the lack of any discussion along these lines is clearly a weakness of the manuscript.

Recommendation

Ask for minor revision

---

## Round 2 · Referee Report · Anonymous (Referee 2) · 2024-5-23

Report

The changes made by the authors have improved the manuscript, clarifying and further discussing several of the points that were vague or ignored in the first version. Therefore, and for the reasons given in my first report (and the addendum to my report after a query from the editor) I recommend publication.

Recommendation

Publish (meets expectations and criteria for this Journal)

---

## Round 2 · Referee Report · Anonymous (Referee 1) · 2024-5-25

Report

The authors have addressed my questions in a satisfactory way, and further improved the paper by including more accurate boundary conditions. I believe the paper should be published in SciPost Physics.

Recommendation

Publish (easily meets expectations and criteria for this Journal; among top 50%)

---

## Round 2 · Author Response

Dear Editor,

First of all we would like to thank the two referees for their careful reading of our manuscript and for raising questions and comments that helped us improve it. We have taken all points into account in our revised version and reply to the points in turn below.

Referee A:

"Apart from a few minor points, detailed below, in my opinion the paper is technically interesting, and timely, as the application of holographic methods to Neutron Star physics has led to important results in recent years. Hence, I believe it should eventually be published in SciPost Physics."

We thank the referee their opinion.

"My only important suggestions for improving the final version are the following: - The authors mention in the introduction, main text, and show in appendix A, that the method used here is basically equivalent to that of Ref [35]. The only two differences are (1) the choice of the $N_c$ scaling in the definition of the isospin chemical potential and the symmetry energy, and (2) the correct inclusion of the abelian spatial components, which were originally ignored in [35] (although the authors might comment on the discussion given in Ref [54] regarding this point, namely the impact of this approximation on the phase diagram of the model). However, when discussing the results for the symmetry energy and proton fraction in Secs 4 and 5, the authors say that they are much closer to the phenomenological values. I believe it should be clarified how much of this improvement actually relies on the conventional $N_c$ scaling, and how much is due to the improvement of the Ansatz for the Abelian gauge fields."

We thank the referee for this question and we have added appendix E addressing this point. Since the $N_c$-scaling just gives an overall factor of $N_c^2=9$ in the symmetry energy, this difference is easy to understand. In order to disentangle the effect of not including the Abelian field in the Ansatz (32) from the $N_c$-scaling, we show the symmetry energy at saturation density as a function of $\lambda$ in Fig. 5, similarly to Fig. 2, but with and without the Abelian field $L$ included (red and green lines, respectively) for $M_{KK}=949$. We observe that neglecting the Abelian field in the Ansatz is justified as a large-$\lambda$ approximation for $\lambda\gtrsim 200$. For $\lambda=16.63$, neglecting the Abelian field $L$, increases the symmetry energy by 46%.

Ref [54] ([55] in the new version of the manuscript) already correctly includes the Abelian spatial gauge field, while it employs a different approximation, that of diagonal $A_i^a$: in the context of that work it is indeed an approximation as quark masses and finite isospin chemical potential can turn on off-diagonal components. However, as explained in the reference, the diagonal approximation becomes exact in the chiral limit and at vanishing $\mu_I$: our work satisfies both criteria since we are computing an expansion around symmetric matter and we neglect the impact of quark masses. Moreover, in the specific limit of vanishing quark masses and isospin chemical potential, the gauge field simplifies even more, needing only the trace component to be turned on. This is the Ansatz we employ and a discussion on deviations from it is included at the end of appendix A.

"- Besides the computation of a more realistic symmetry energy and proton fraction (and especially if the improvement is mostly due to the $N_c$ scaling), I believe the authors should clarify what is the main advantage of carrying out the computation in this way. For instance, what is the impact for future applications to Neutron Stars? How would those hypothetical results compare with those from other holographic NS models, such as those based on VQCD? What other applications would benefit from the quantization analysis?"

In the end of Sec. 3, we comment on the advantages of our way of carrying out the computation of the symmetry energy, which are a compact integral formula, no $N_c$-scaling ambiguities, and facilitation of identifying the correct form of the Chern-Simons action in the bulk in the presence of discontinuous fields ($H$).

"- In the discussion section, it is said that the holographic popcorn transition is "already taken into account", but this is the first mention of it. I believe this comment is too vague. If it is part of the computation, it should be explained in the main text when it enters, and it's impact should be discussed, as well as it's comparison with non-holographic models, quarkyonic matter, etc."

We have added an improved discussion of the popcorn transition in the discussion and removed the previous remark, that was speculating in interpreting the behavior of the symmetry energy around saturation density in the HW model, which turned out to be a computational inaccuracy that we have fixed in the revised version of the paper.

"Besides addressing the questions raised in the above report, here is a list of minor points the authors should consider: - At the beginning of section 2, it is said that at low energies both models are described by gauge theories on AdS5, but then the metric for the WSS model is different. I understand what the authors mean, but the presentation seems a bit confusing."

We have corrected this by writing "(AdS$_5$-like for the WSS model)" above Eq. (2).

"- The use of the coordinate "z" in both cases has its advantages, but it is a bit confusing that in the HW case there is a scale "L" for the AdS radius, but in the WSS model "z" is dimensionless. Perhaps this could be clarified as well."

We apologize for the confusion and have fixed this by using dimensionless quantities in both models setting $L=1$. See the comments under (4) and the correction in (9).

"- In footnote 3 one could also consider citing recent papers where the effect of the quark mass on the phase diagram of the system was considered."

We have added some references in footnote 2: we cite three recent works within the WSS model [53,54,55] ([55] being already present in the earlier version, albeit not in this specific footnote), and one work within the HW approach [56].

"- Below Eqs (35)-(38) it would be useful to explain more carefully (at least in words) why some terms quadratic in $\chi$ are important while others are ignored."

We have added a comment under (38), explaining why the inclusion of the $\chi^2$ terms in the fields $H$ and $\hat{a}_0$ would lead to an order $\chi^4$ effect for the symmetry energy.

"- It would be interesting to discuss in more detail what happens with the equivalence discussed in App. A for larger values of the isospin chemical potential, and how it could affect studies of the phase diagram and of Neutron Stars within these models."

We have added a discussion at the end of App. A about the larger values of the isospin chemical potential.

"- Given the equivalence with previous works discussed briefly in the main text and in detail in appendix A (up to some details) the paper would benefit from a more explicit discussion of the motivation for carrying out the present computation, not only from the technical point of view but also in view of possible future directions of investigation."

We have added general comments on this at the end of Sec. 3.

Referee B:

"The paper applies two holographic models to compute the symmetry energy of nuclear matter. In both models a simple approximation of dense, homogeneous nuclear matter is used. The questions addressed in the paper are timely and the results are new (improving on existing results in the literature). The paper is well written, all calculations are explained in a clear and transparent way. I do not have any major concerns and thus recommend publication in principle."

We thank the referee for the comments.

"I have some comments that should be address before publication: (1) In the abstract, the authors claim that after adjusting the parameters the results "pass all experimental constraints". Isn't this a bit too strong? For instance, the authors say themselves in the paper that the saturation density is not reproduced correctly (and they never mention additional properties of nuclear matter such as the binding energy or the compressibility). Additionally, they also do discuss in the paper that mesonic properties and the nucleon mass are not correctly reproduced (with the new parameters used here). All this makes the formulation in the abstract very misleading and I would strongly suggest to find a more honest formulation."

We thank the referee for this comment. We intended only that the symmetry energy could pass the experimental constraints in the measured range $\approx[0.5,2]\times\rho_0$. We have removed the part of the phrase that was misleading and now the sentence in the abstract reads "We find that the SE can be well described in the WSS model if we allow for a larger 't Hooft coupling and lower Kaluza-Klein scale than is normally used in phenomenological fits."

"(2) Somewhat related to the previous point: The authors say on p10 that the saturation density $rho_0$ is a factor 3 too large. But this can only be true for a fixed parameter set. It would be good to know what that value is in the regime where the symmetry energy assumes realistic values. Also, $rho_0$ probably changes from curve to curve in fig 1 for example. Some warning or a comment on the range of values $rho_0$ can assume might be helpful.

We thank the referee for this comment. We have included the saturation density for the WSS model with the mesonic fit in (50), which is about 2.9 times too large; and also the saturation density for the HW model in (51), which is only about 22\% too large. As suggested by the referee, we have also shown the saturation density for the WSS model with the different fit having $\lambda=60$ and $M_{KK}\sim 390\MeV$ in (53); it is only about 10\% too large, but depends drastically on $M_{KK}$, so it can easily be improved but at the cost of getting into tension with the neutron skin thickness constraint at higher densities, see the comments in red in Sec. 4. The variation of the saturation density in these formulae depend cubically on $M_{KK}$ (or $L^{-1}$) and hence vary strongly with the scale of the model.

"(3) A more general comment about the interpretation of the results: All numerical results in Sec 4 and 5 are presented more or less as observations without trying to interpret or understand them. For instance, do the results suggest to keep using the models because the symmetry energy comes out right, despite shortcomings for $rho_0$ (and possibly the never mentioned binding energy)? Or should the models or the approximations for nuclear matter be improved? Are the deviations and (partial) agreement with real-world physics expected and can they be explained for instance with large-N_c artifacts?

As better presented in the revised version of the paper, we think that the improved fit (which is also similar to that of [59]) makes it possible to have a correct symmetry energy, correct saturation density and better (smaller) nuclear mass, at the expense of sacrificing the mesonic sector of the model. As for the nucleon mass, its value in the setting of the homogeneous Ansatz may not be important as the Ansatz knows nothing about the nucleon, but similar mechanisms lead to the fact that the chemical potential at saturation density is somewhat too large (about 1200 MeV) as compared with the nucleon mass. The fact that the meson sector is not fitted well with the same parameter set that fits the baryonic sector well, is a typical problem of large-$N_c$ theories, since the meson is $\bar{q}q$ and the baryon has $N_c>>1$ quarks. If one should have any hope for fitting both the mesonic sector as well as the baryonic sector of the model, one would at least be forced to include difficult $1/N_c$ corrections.

We have also added a comment on improvements on the homogeneous Ansatz at the end of Sec. 6, that may be taken into account in future studies; that is, we have worked exclusively with the discontinuity of the field $H$ at the IR end, but this discontinuity could in principle move closer to the boundary, as dictated by minimization of the action.

Not to mention also the trivial point of the homogeneous Ansatz is not expected to work well when reaching saturation density or lower densities; some structure is expected to appear, but would require more difficult calculations which we leave for future studies.

"Is there a physical explanation why a larger 't Hooft coupling leads to a more realistic symmetry energy? Given that the paper also contains interesting theoretical novelties it might be acceptable to keep these interpretations short, but the lack of any discussion along these lines is clearly a weakness of the manuscript."

We thank the referee for giving us the opportunity to expand the discussion on the topic. We have added a discussion on the larger 't Hooft coupling at the end of section 4. In particular, we show how the single baryon picture already suggests that such a limit is relevant to obtain a system that is well approximated by its classical description, with small quantum corrections appearing after moduli space quantization. At the same time we observe that such intuition is then confirmed by the improvement of many observables at the same time, leading to a baryonic sector which is closer to phenomenology.

On top of the changes suggested by the referees, we have also implemented a minor improvement of the boundary conditions (39), which has lowered the symmetry energy about 10-20%. This boundary condition comes from varying the entire action, including taking into account the total derivative term that comes about when computing the Euler-Lagrange equations, see [57] for more details.

We have taken into account all points and suggestions made by the two referees and hope that the paper can now be considered for publication in SciPost Physics.

Sincerely, the authors

---

## Round 2 · List of Changes

Summary of changes (all corrections are displayed in the manuscript
with red colors):

-abstract improved (misleading phrase removed).

-AdS$_5$ changed to AdS$_5$-like on pp.3.

-$L$ set equal to 1 after (4) and in (9) and comments added: now $z$
is dimensionless in both HW and WSS.

-pp.5, footnote 2: references on phase diagram with quark mass added.

-comment on leading-order in $\chi^2$ being sufficient for the
symmetry energy computation added after (38) on page 7 and comments
regarding finite isospin chemical potential added at the end of
Appendix A on pp.17.

-improved boundary conditions employed in (39) and a comment and
footnote 4 explaining its reason and impact is given immediately
after (39).

-Advantages of the method are discussed at the end of Sec. 3 on
pp.8-9.

-Symmetry energy expansion parameters updated and given at the chosen
scale with explicit KK scale dependence for WSS and HW in (48),(49)
as well as (52) for the new improved fit.

-The physical saturation density for WSS and HW is given in (50),(51)
with explicit KK scale dependence and in (53) for the new improved
fit.

-Comment on interpretation of why a larger 't Hooft coupling provides
more physical observables added on pp.12-13.

-Discussion on popcorn transition improved at the end of Sec. 6 on
pp.15.

-Appendix E added explaining how much of the improvement of the
symmetry energy is due to the inclusion of the Abelian field $L$ and
how it becomes a good approximation in the large 't Hooft coupling
limit.

-References [53,54,56,58,62] added.

---

## Editorial Decision

published